# Therapeutic Approaches in Adult Primary Spinal Cord Astrocytoma: A Systematic Review

**DOI:** 10.3390/cancers14051292

**Published:** 2022-03-02

**Authors:** Elena Anghileri, Morgan Broggi, Elio Mazzapicchi, Mariangela Farinotti, Andrea Botturi, Irene Tramacere, Marcello Marchetti

**Affiliations:** 1Unit of Molecular Neuro-Oncology, Fondazione IRCCS Istituto Neurologico Carlo Besta, 20133 Milan, Italy; 2Department of Neurosurgery, Fondazione IRCCS Istituto Neurologico Carlo Besta, 20133 Milan, Italy; morgan.broggi@istituto-besta.it (M.B.); elio.mazzapicchi@unimi.it (E.M.); 3Neuroepidemiology Unit, Scientific Directorate, Fondazione IRCCS Istituto Neurologico Carlo Besta, 20133 Milan, Italy; mariangela.farinotti@istituto-besta.it; 4Neuro-Oncology Unit, Department of Clinical Neurosciences, Fondazione IRCCS Istituto Neurologico Carlo Besta, 20133 Milan, Italy; andrea.botturi@istituto-besta.it; 5Department of Research and Clinical Development, Scientific Directorate, Fondazione IRCCS Istituto Neurologico Carlo Besta, 20133 Milan, Italy; irene.tramacere@istituto-besta.it; 6Unit of Radiotherapy, Department of Neurosurgery, Fondazione IRCCS Istituto Neurologico Carlo Besta, 20133 Milan, Italy; marcello.marchetti@istituto-besta.it

**Keywords:** astrocytoma, spinal cord, primary, therapy, outcome, safety, spinal cord astrocytoma (SCA), adult, outcome

## Abstract

**Simple Summary:**

Adult primary spinal cord astrocytoma (SCA) is a very rare disease, with no standardized consensus about its therapy. We focus on treatment efficacy based on systematic review: only 18 papers were eligible for the analysis, including 285 patients. No clinical trials results were available. Not enough data were extracted to determine a univocal treatment plan for SCA. Given the rarity of these diseases, a collaboration among institutions is mandatory to establish a standard for study conduction (homogenous inclusion criteria and method of analysis), to perform homogenous studies and define future evidence-based recommendation. Contextually, multicentric clinical trials with molecular investigations are strongly advised to better manage SCA and unveil their biology.

**Abstract:**

The issue: Gliomas are primary tumors arising from supporting cells of the central nervous system (CNS), usually in the brain. The 2021 World Health Organization (WHO) classifies gliomas as adult-type diffuse gliomas or circumscribed astrocytic gliomas depending on their histology and molecular features. Spinal astrocytic gliomas are very rare, and nowadays no standard of therapy is available. Treatment options are limited: surgery is often not radical, and adjuvant therapies include mostly radiotherapy (RT) or systemic chemotherapy (CHT). There is lack of knowledge about the efficacy and safety of therapies and their multidisciplinary approaches. The aim of the review: A systematic review of the literature from January 2000 to June 2021 was performed, including both clinical trials and observational studies on histological adult primary spinal cord astrocytomas (SCA), with a minimum follow-up of 6 months and reporting the overall survival, progression-free survival or clinical neurological outcome after any therapeutic approach (surgery, RT or CHT). What are the main findings? A total of 1197 citations were identified by the Medline search and additional records; based on our inclusion criteria, 18 studies were included with a total of 285 adult patients. We documented the lack of any clinical trial. What are the conclusions? The available literature data are limited to series/retrospective studies, including heterogeneous patients, i.e., astrocytoma as well as ependymoma or pediatric/adult age, with scanty data on the outcomes of interest. No clinical trials have been run. Due to the rarity of this disease, multicentric clinical trials with molecular investigations are mandatory to better manage such a rare disease.

## 1. Introduction

Primary spinal cord tumors are rare, representing about 4–8% of all tumors of the central nervous system (CNS) [1], that is, 10 to 15 times less common than their cranial counterparts. They are conventionally divided according to their anatomic location into three categories: extradural, intradural extramedullary, and intramedullary. Intramedullary spinal cord tumors (IMSCTs) are the rarest type, representing 20% of all intraspinal tumors. Their incidence has been reported as 0.22 per 100,000 [2]. Spinal cord gliomas in adults include ependymomas and astrocytomas, representing 90% of IMCSTs. Ependymomas account for 60%; spinal cord astrocytomas (SCA) account for 30% of IMCSTs and only 3% of CNS astrocytomas [3]. The remaining IMCSTs include hemangioblastomas (3 to 8%), metastases (2%), primary CNS lymphomas and miscellaneous tumors.

Given their rarity, deep knowledge of these tumors is still lacking, and nowadays therapy is mostly derived from the brain counterpart, but “standard of care” or homogenous guidelines have not been developed yet.

Spinal Magnetic Resonance Imaging (MRI) is the state of the art of diagnosing intradural spinal tumors. Sometimes an additional Computed Tomography (CT) scan and neurophysiological examination can be valuable to plan surgery. The management of these lesions depends not only on the histopathological diagnosis but also on the clinical presentation and the anatomical location, allowing radical or less invasive surgery, also supported by intraoperative neuromonitoring, with minimally invasive approaches currently available. 

While surgery is the mainstay treatment for grade 1 astrocytoma, and to a lesser extent for higher infiltrative-grade lesions [4], radiotherapy (RT) and chemotherapy (CHT) are of growing importance after partial removal and in recurrent and multifocal lesions. Besides the traditional alkylant drugs, targeted therapy could represent incoming promising therapy based on targetable gene mutations [5].

The aim of this review is to evaluate treatments’ effectiveness and safety, collecting all the English literature about management of adult SCA from January 2000 to June 2021. Starting from the existing evidence, we aim to formulate recommendations to better address and understand the disease.

## 2. Materials and Methods

A systematic literature search concerning the management of SCA was performed and the Preferred Reporting Items for Systematic Reviews and Meta-Analyses (PRISMA (The review was not registered.)) statement [6] was used to report the results. 

### 2.1. Search Strategy

A comprehensive electronic literature search was performed in PubMed from January 2000 to June 2021. Combinations of the following subject headings and keywords were used across all databases: ((astrocytoma [MeSH Terms]) OR (glioblastoma (GBM) [MeSH Terms])) AND ((spine [MeSH Terms]) OR (spinal cord [MeSH Terms])) AND (2001/1/1:2021/06/30 [pdat]) AND (((radiotherapy) OR (radiosurgery)) OR (surgery) OR (drug therapy)). Further filters as “human”, “adult” and “English” were applied.

The reference lists of all included studies and relevant review articles were manually searched to identify any new eligible study.

### 2.2. Eligibility Criteria

Studies were included if they reported on:Adult patients (≥18 years old);Histologically confirmed cases with primary spinal location of glioma, glioblastoma, glial tumor, astrocytoma;At least one of the following outcomes, after any treatment:
Median Overall Survival (OS), 3-month, 6-, 9-, or 1-year OS;Progression Free survival (PFS) (median PFS or 6-month PFS or longer); Clinical neurological outcome (including modified McCormick scale (MMS) [7] or Karnofsky Performance Status (KPS));
At least 6-month follow-up;Clinical trials, observational or case series studies.Studies were excluded if they:Studied recurrent or secondary spinal glioma (developed from primary cerebral glioma);Included overall glioma (astrocytoma and ependymoma) and data on the subgroup were not available;Exclusively investigated pediatric cases;Were not published in English language;Were letters, conference papers, review or case report articles, or part of a book.

### 2.3. Data Collection

The title and abstract of all identified publications were screened to assess suitability for inclusion by two authors (EA, MB). Publications considered potentially eligible were read in full by two authors (EA, EM), and a third (MB) was involved when consensus on the study inclusion was not achieved.

Papers including other spinal gliomas subtype (e.g., ependymoma) or pediatric cases were exclusively included if the specific data required for the present review were specified.

For each included study, data were extracted by one author (EA) and checked by another (MB). 

Collected data were on age group, sex, sample size, treatment methods (treatment protocol, dosages, cycles). The main radiological features (including segment site, length extension), histological details (astrocytoma grade, mitotic index) and, if available, molecular features were also collected. 

The primary endpoints of interest were the following: OS, PFS or clinical neurological outcome.

Secondary endpoints were adverse events after therapies.

### 2.4. Risk-of-Bias Assessment

Risk of bias was evaluated according to Munn et al. 2020, based on a 10-item assessment (Table 1).

The system was developed to evaluate different risks of bias and the completeness of information in case series studies. Questions 1, 4, and 5 can be considered signaling questions for the “bias in selection of participants into the study”; questions 2 and 3 for the “bias in measurement of outcomes”; questions 6 and 7 for the “bias in selection of the reported results”; and question 8 for the “bias due to missing data”. The answer, “yes,” indicated a low risk of bias, while a negative answer, “no,” a high risk. Unclear (NC) or not applicable (NA) were specific answer option.

The quality of evidence of the selected papers was analyzed by two independent authors (EA, EM).

### 2.5. Statistical Analysis

For our study, we planned to perform a meta-analysis. However, due to the limited number of studies and scarce available data, the meta-analysis was not feasible. Accordingly, results were synthesized qualitatively.

## 3. Results

### 3.1. Search Selection

The PubMed search identified 1195 citations published between January 2000 and June 2021; 522 papers were selected after applying the filters “English”, “human” and “adult”. Two additional records were identified by manual searching of the bibliography of relevant papers and reviews on the subject. After reading titles and abstracts, we retained 40 potentially eligible articles. Among these, 18 studies [9,10,11,12,13,14,15,16,17,18,19,20,21,22,23,24,25,26] were included in the review (Figure 1).

Appendix A summarizes the characteristics of the included studies. Five studies were observational, and the remaining 13 studies were case series. Notably, no clinical trials were published.

### 3.2. Epidemiology, Clinical Presentation, and Diagnosis

A total of 285 adult patients with SCA were identified.

The age range included the entire adulthood; when declared, the male sex ranged from 33% (*n* = 1/3 [22]) to 84.6% (*n* = 11/13 [23]) of the series. As shown in Appendix A, the gliomas spinal site in cervical thoracic and lumbar regions ranged from 0% (*n* = 0/3 [17]) to 70% (*n* = 7/10 [25]), from 20% (*n* = 2/10 [25]) to 77% (*n* = 10/13 [10]) and from 0 (*n* = 0/13 [10]; *n* = 0/24 [11]; *n* = 0/22 [15], *n* = 0/6 [18]; *n* = 0/5 [19]; *n* = 0/25 [21]; *n* = 0/6 [22]; *n* = 0/13 [23]; *n* = 0/21 [26]) to 41% (*n* = 5/12 [14]), respectively. The lesion involved more than one region from 0% cases [10,15,19,22,24,25,26] to 43% (*n* = 10/24 [11]). When described, the median segment extension ranged from 2.3 segments [10] to 4 [11,17]. Signs of dissemination are reported to be present in 2/16 papers [11,15]. Fifteen out of eighteen papers investigated the additional presence of brain localization during disease evolution and did not describe any type of cerebral involvement.

Clinical data were reported only in few papers, often using no objective scale [11,13,14,18,20,21,22,25,26]. Median pre-surgical MMS was described in seven reports [10,11,15,16,19,25], and ranged from 1 (*n* = 10 [25]) to 3 (*n* = 5 [19]). Only one report described pre-surgical KPS estimated as 80 (*n* = 13 [10]).

Concerning the tumors histological grade according to the WHO classifications of tumors of CNS (edition from 2000 to 2016), studies included Grade I SCA from 0% [9,10,11,15,17,18,19,22,23,24] till 100% (*n* = 10 [25]); grade 2 from 0% [11,15,18,19,22,23,25] to 100% (*n* = 13 [10]); grade 3 from 0% [10,13,18,19,22,25] to 87% (*n* = 21/24 [11]); and grade IV from 0% [10,13,25] to 100% (*n* = 6/6 [18]; *n* = 5/5 [19]; *n* = 6/6 [22]). 

Very few molecular data were available. All the high-grade glioma investigated for IDH1 mutation were negative (*n* = 45 [19,22,23,24]) contrasting with the grade 1 glioma (*n* = 15) harboring IDH1 mutation in all cases based on Jiang 2020 [25].

Limited to one series [25] focusing on 10 spinal grade I pilocytic glioma, 0 out of 4 tested were BRAF V600 positive, 8/10 tested had p53 mutation and 1/4 tested was H3K27 positive.

In one series of *n* = 13 grade III-IV SCA, 0 out of 13 tested had BRAF V600E mutation or IDH1 mutation at immunohistochemistry (IHC); 3/10 tested had TERT promotor mutation, 1/10 EGFR mutation, 8/10 TP 53 mutation and 6/13 H3K27M mutation, all mutations detected by Next Generation Sequencing (NGS). In addition to the most common glioma mutations, others were detected as PPM1D mutation (3 out 10 tested samples), PIK3CA mutation (2/10 tested), BCOR mutation (1/10 tested) and SETD2 mutation in 1/10 test SCA [23]. 

Two other papers on high-grade glioma reported TP53 mutation in 1/2 tested cases and in 2/6 tested cases, respectively [19,22].

### 3.3. Therapy

Therapeutic approaches included surgery, CHT and RT. As mentioned above, neither clinical trials nor comparative studies were available.

Appendix A reports the details of therapies. In particular, surgical approaches included biopsy, partial resection or gross total resection. Biopsy was performed from 0% [11,13,15,21,22,25] to 61% (*n* = 8/13 [23]) of the cases; partial resection from 13% (*n* = 3/23 [12]) to 100% (*n* = 6/6, grade IV SCA [22]) and gross total resection ranged from 0% [10,17,18,20,22] to 80% (*n* = 20/25 [21]; and *n* = 8/10 grade I SCA [25]).

RT plans details were lacking in 9 studies out 15; when reported (in six studies with *n* = 59 treated patients), RT were mostly conformational 3D, with total Gy dose ranging from 40 to 56 Gy.

Overall, 46.1% of patients (*n* = 94/204) received systemic therapy. Regarding CHT protocols, the majority of drugs were alkylants (mostly temozolomide (TMZ); Carboplatin, Cyclophosphamide or Carmustine/Lomustine) or Vincristina. Xiao et al., 2016 and Yanamadala et al., 2016 also reported Bevacizumab [20,22]. Alvi et al., 2019 reported 1 case (out of 13) treated with Panobinostat after RT, alive at 15-month follow-up after subtotal resection for grade IV thoracic (T9–T12) glioma and one case treated with immunotherapy (any further details are not available) [23].

Based on the incomplete data reported in the reviewed papers, it was not possible to specify if and when CHT was combined or adjuvant to RT.

### 3.4. Outcome

Table 2 shows the outcomes reported in the included studies.

The median PFS were reported in six papers, ranging from 7 to 138.8 months [17,18,20,21,24,25]. The only paper that investigated 5 years PFS reported 93% in a 13-sample series of grade 2 glioma [10].

Eight out of eighteen papers reported median OS. In one study, investigating 13 gliomas, median OS was not reached at 15 years of follow up, but 77% of cases were grade 1–2 [20]. Other seven papers reported median OS ranging from 12 to 43.3 months [9,11,12,15,18,23,24]. 

One-year OS was reported as 100% (focusing on grade IV SCA [22]), 46% (grade III–IV SCA [23]), 82.3% (grade II–IV SCA [24]).

Two-year OS was reported as 20% [19] and 50% [23]; both series included high grade gliomas only.

Five-year OS was reported as 100% by Robinson et al., 2005 (all grade II glioma cases) [10], 83% by Xiao et al., 2016 (any grade SCA) [20], and 67% by Seaman et al., 2021 (any grade SCA) [26].

Regarding surgery, five papers reported pre-surgical and relative post-surgical MMS: in two studies no MMS changes were reported [19,25]; the other three described post-surgical MMS worsening [10,11,15].

As reported by the literature, the infiltration of SCA in the perilesional parenchyma, usually visible on MRI scans, makes it difficult to achieve complete surgical resection, differently from spinal ependymomas [26]. In our review, only 25% of cases underwent to total surgical removal, and no prognostic role was investigated.

No studies reported on therapy-adverse events. 

### 3.5. Risk of Bias Assessment

Results of bias scoring based on recent work of Munn et al., 2020 [8] is outlined in Table 3. 

Only one study received the maximum total score of 10 [20], indicating the lowest risk of bias. The lowest score was 1/10 (including items 4 and 5, unclear, and item 10, not applicable) given to Kim et al., 2011 indicating the highest risk of bias [17].

### 3.6. Prognostic Factors

The heterogeneity of the reviewed studies, as well as biases due to their uncontrolled designs, preclude us from making any meaningful conclusions with regard to the therapeutic approaches and/or potentially prognostic factors, if any.

Among 28 high-grade gliomas, the relation between survival and age, gender, tumor grade, treatment modality, extent of resection (EOR), p53 expression, and MIB-1 LI were analyzed by Kaplan–Meier curves and the log-rank test: only age higher than 40 years resulted as significant negative prognostic factor [9].

Older age at surgery was the only significant variable that predicted decreased probability of PFS in a series (*n* = 13) including any grade of gliomas [20]. In the same work, other variables associated with four outcomes (improved neurological status, local control, PFS, and OS) were analyzed by multivariable logistic and Cox proportional hazards regression models, resulting in lack of any association [20].

Others reported tumor grade as the only independent significant factor on multivariate analysis of PFS in *n* = 25/26 samples [21]. In the same study, the outcome (both OS and PFS) of low-grade astrocytoma was significantly better than high-grade astrocytoma. The univariate analysis identified tumor grade and tumor location as being significantly associated with OS. Other factors significantly associated with OS on univariate analysis were preoperative neurological status, postoperative neurological status and surgical extent. The Ki-67 index score was a significant negative predictor of OS: however, the data included only 15/26 patients, and resulted as having low statistical power. Tumor grade and location were the only variables that remained independently significant in the multivariate analysis. Postoperative RT (performed only in patients with high-grade tumors) did not impact outcome. Prognostic factors of PFS were tumor grade, surgical extent, and Ki-67 index, analyzed in the univariate Cox hazards models [21].

Two other studies specifically addressed the role of molecular data on astrocytoma prognosis [23,24]. H3.3 K27M mutation was reported as significantly associated with better outcome [23,24].

Seong et al., 2018 [24] showed H3.3 K27M mutation as a positive prognostic variable (both PFS and OS) in *n* = 25 cases of grade IV spinal gliomas. The other analyzed variables, including gender, EOR, RT, chemotherapy, and resulted in no prognostic factors [24].

H3K27Mmutant tumors were associated with longer overall survival in 13 high-grade infiltrating gliomas, by Kaplan–Meier curves and the log-rank test [23]; on pair-wise comparison, no significant difference in the overall survival was found between patients with H3K27Mmutant and H3K27M-wildtype tumors (or between H3K27M-mutant and TERT promoter-mutant tumors). Notably, patients with TERT promoter mutant high-grade infiltrating gliomas of the spinal cord had significantly lower overall survival compared to those with TERT wild-type tumors—a group that includes all H3K27M-mutant cases [23]. The low number of cases undermine the statistical power of these analyses and any results needs to be confirmed.

## 4. Discussion

Intramedullary astrocytomas represent a rare neoplasm constituting approximately 2–4% of all primary CNS tumors, and are even rarer in adulthood. Literature is very scarce on SCA biology as well as their evolution and guidelines on therapies are lacking. The reported survival outcomes of adult patients with these neoplasms are very heterogeneous, and the majority limited to the early post-surgical course [27]. Based on the tumor location, neurological signs often deeply impact patients’ quality of life. Clinical evaluation is a key point of the overall evaluation, also including the pain component [4,15,25].

In contrast to their intracranial counterpart, the primary spinal astrocytomas still have no management consensus among clinicians but they cause even more disabling outcome and also affect younger subjects than the intracranial cases. Considering treatments, less well-defined margins between the tumor and normal spinal cord make gross total resection an extremely great challenge, and the effect of CHT, which like TMZ was proven to be effective for intracranial GBM, remains controversial on spinal cord GBM [28]. Treatments were most frequently surgery and RT. In some studies, adjuvant CHT (mostly alkylant-based) was used. 

All the included studies had the well-known limitations of retrospective studies; besides, the selected papers were from 2000 to nowadays, a 20-year time period in which radiological, neurosurgical and adjuvant chemotherapies improvements could potentially affect the outcome differently in different years. Therefore, the extracted data for the present review did not allow us to determine any statements about the planned endpoints.

Some of the included studies attempted to identify prognostic factors but faced the limitation of their study design [9,20,21,23,24]. In particular, these studies—although applied multivariate models—did not obtain reliable estimates based on their small sample sizes.

Some other authors investigated prognostic factors for spinal gliomas, but including heterogeneous case series. A not included retrospective study with 83 pediatric and adult grade I–IV gliomas [29] showed that higher WHO grade among all patients (not reported adults vs. pediatric data) was associated with worse OS (*p* < 0.0001) and PFS (*p* = 0.0003). Among patients with infiltrative tumors, neither EOR nor RT was associated with a difference in outcomes in multivariate analysis; however, among patients with infiltrative astrocytomas, CHT was significantly associated with improved PFS (hazard ratio = 0.22, *p* = 0.0075) but not OS (hazard ratio = 0.89, *p* = 0.83) at multivariate analysis [29].

This systematic review had several limitations. Due to the small number of included patients and scarce available data, the meta-analysis was not feasible. 

We underline that spinal astrocytoma molecular features, are still largely under-investigated, and that they could result extremely useful for therapeutic approach (i.e., BRAF–MEK inhibitors for BRAF mutant glioma). 

As consequence of the difficulties of surgical exeresis, histologic grading can be challenging in spinal cord astrocytomas because of the relatively small samples obtained with surgical procedure. Therefore, grade-defining molecular biomarkers would be particularly useful for the accurate diagnostic classification of these tumors [30].

The few data available in the literature reported that molecular profile in spinal gliomas did not mirror the cerebral counterpart [31].

In 2016, Shankar et al. reported a statistically significant difference (*p* < 0.001) comparing H3F3A K27M presence in grade III and grade IV vs. grade I and grade II (mostly pediatric) astrocytomas. The most recurrent findings in grade I spinal cord astrocytomas were a BRAF-KIAA1549 translocation (*n* = 3/10) and BRAF copy number gain (*n* = 5/10). WHO grade II astrocytomas were similarly characterized by alterations involved in the MAPK-ERK or PI3K pathways, including BRAF-KIAA1549 translocation (*n* = 1/3) and BRAF amplification (*n* = 2/3) [32]. BRAF fusion detection in grade I and grade II SCA was also confirmed by Lebrun et al., 2020. All of the grade III and grade IV glioma presented at least one molecular alteration, with the most frequent one being the H3F3A p.K27M mutation. The H3F3A p.K27M mutation showed a better prognosis [31]. The combination of retained H3K27me3 and negative EZH2 expression was also reported as being related to favorable overall survival (*p* = 0.03) among WHO grade II-IV cases by another group [33]. 

Nagashima et al., 2021 focused on the impact of Driver Genetic Mutations in Spinal Cord Gliomas and concluded that gliomas with H3F3A mutations were associated with accelerated tumor-associated spinal cord injury, leading to functional impairment. Conversely, the presence of IDH mutations, which are rarely reported in spinal gliomas, indicated a relatively favorable functional prognosis [34]. Biczok et al., 2021 identified five distinct subgroups in 26 patients (adults and pediatrics) of spinal astrocytomas based on molecular data. Histology and NGS allowed the distinction of five tumor subgroups: glioblastoma IDH wildtype (GBM); diffuse midline glioma H3 K27M-mutated (DMG-H3); high-grade astrocytoma with piloid features (HAP); diffuse astrocytoma IDH mutated (DA), diffuse leptomeningeal glioneuronal tumors (DGLN) and pilocytic astrocytoma (PA). Within all tumor entities GBM (median OS: 5.5 months), DMG-H3 (median OS: 13 months) and HAP (median OS: 8 months) showed a fatal prognosis. HAP are characterized by CDKN2A/B deletion and ATRX mutation. 50% (*n* = 4/8) of PA tumors carried a mutation in the PIK3CA gene, seemingly associated with better outcome [5]. All of them represent a targetable mutation.

## 5. Conclusions

Based on the rarity of adult SCA and the limited number of studies available, the literature did not provide enough data to determine a recommended treatment plan for SCA. 

Given the rarity of this disease, a collaborative effort among institutions establishing a standard line for study conduction (homogenous inclusion criteria, statistical analysis and data presentation) could provide a large number of studies directly comparable to one another in the future. This will allow us to define evidence-based recommendations. 

Ideally, studies designed to best answer the endpoint question will include a large multicenter cohort of patients stratified on factors including age, functional status, histological grade, EOR, and RT modality to minimize confounders and allow comparisons. 

## Figures and Tables

**Figure 1 cancers-14-01292-f001:**
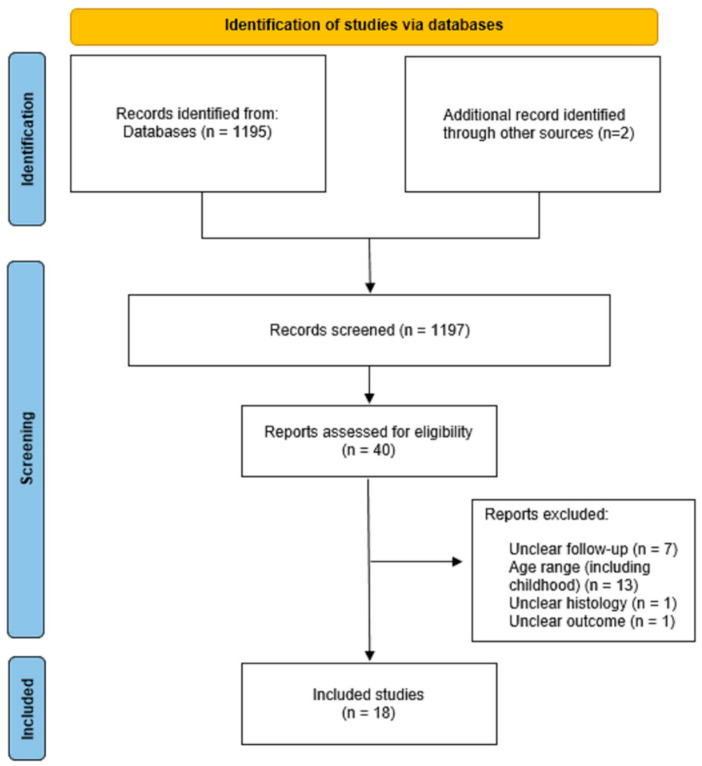
Flow diagram for systematic review.

**Table 1 cancers-14-01292-t001:** Risk-of-bias assessment—Items [8].

Question	Yes	No	Unclear	Not Applicable
1. Were there clear criteria for inclusion in the case series?	O	O	O	O
2. Was the condition measured in a standard, reliable way for all participants included in the case series?	O	O	O	O
3. Were valid methods used for identification of the condition for all participants included in the case series?	O	O	O	O
4. Did the case series have consecutive inclusion of participants?	O	O	O	O
5. Did the case series have complete inclusion of participants?	O	O	O	O
6. Was there clear reporting of the demographics of the participants in the study?	O	O	O	O
7. Was there clear reporting of clinical information of the participants?	O	O	O	O
8. Were the outcomes or follow-up results of cases clearly reported?	O	O	O	O
9. Was there clear reporting of the presenting sites’/clinics demographic information?	O	O	O	O
10. Was statistical analysis appropriate?	O	O	O	O

**Table 2 cancers-14-01292-t002:** The survival outcomes (as OS or PFS). Abbreviations: PFS = Progression Free Survival; OS = Overall Survival; Control disease: SD = Stable Disease, PR = Partial Response; PD = Progression Disease; NR = Not Reached; LC = Local Control; FU = Follow-Up; yr = year. Empty cell = Data Not Available. (^a^): overall OS (total series of 36 cases) (^b^): overall PFS (total series of 16 cases, including 6 pediatric).

		PFS from First Diagnosis				PFS from Recurrence		Overall Survival (OS)			
Reference	Sample Size	PFS (Months)	12 Months PFS (%)	Other PFS Endpoint	Local Control	24 Months PFS (%)	Control after First Recurrence	Median OS (Months)	1-Year OS (%)	2-Year OS (%)	Other OS Endpoint
Santi M [9]	28							33 (24–42) ^a^			
Robinson CG [10]	13			5-yr = 93%							5-yr = 100%
				10-yr = 80%							10-yr = 100%
				20-yr = 60%							20-yr = 75%
McGirt M [11]	24										
Nakamura M [12]	23										
Erdes C [13]	15										
Matsuyama Y [14]	12										
Raco A [15]	22										
Karikari I [16]	21	62.3									
Kim WH [17]	3	7.0			83% SD 17% PD		34% PR	NR			
							50% SD				
							17% PD				
Chamberlain MC [18]	6										
Liu A [19]	5										
Xiao R [20]	13	66				5-yr = 63%	5-yr-LC = 83%	NR (15-yr FU)			5-yr = 83%
Ryu SJ [21]	25	Grade I–II = 138.8 (12–480);								Grade I–II = 56.4 (28–480);	
		Grade III–IV = 6.6 (1–13)								Grade III–IV = 12 (1–36)	
Yanamadala V [22]	6		100						100		
Alvi MA [23]	13							13	46	50	
Yi S [24]	25	18.5						37.1	82.3		
Jiang Y [25]	10	19 (9–108) ^b^			40% PD						
Seaman SC [26]	21										5-yr = 67%

**Table 3 cancers-14-01292-t003:** Risk-of-bias assessment based on the 10-items scale by Munn et al., 2020 [8]. Abbreviations: NC = Not Clear; NA = Not Applicable.

Risk-of-Bias Assessment-ITEMS [8]											
References	1	2	3	4	5	6	7	8	9	10	Total Score
[9]	yes	No	yes	NC	yes	yes	no	yes	no	NA	5
[10]	yes	Yes	yes	NC	yes	yes	yes	yes	yes	NA	8
[11]	yes	Yes	yes	Yes	yes	yes	yes	yes	NA	yes	9
[12]	no	No	yes	Yes	no	no	no	yes	NA	yes	4
[13]	no	Yes	yes	Yes	yes	yes	yes	yes	NA	yes	8
[14]	no	yes	NC	Yes	yes	yes	no	yes	NA	no	5
[15]	no	no	yes	Yes	yes	no	yes	yes	NA	yes	6
[16]	no	yes	yes	Yes	yes	yes	no	yes	NA	yes	7
[17]	no	no	yes	NC	NC	no	no	no	no	NA	1
[18]	yes	yes	yes	Yes	yes	yes	no	yes	no	NA	7
[19]	yes	yes	yes	NC	yes	yes	yes	yes	no	NA	7
[20]	yes	yes	yes	Yes	yes	yes	yes	yes	yes	yes	10
[21]	yes	yes	yes	NC	yes	yes	yes	yes	yes	yes	9
[22]	yes	yes	no	NC	yes	yes	yes	yes	no	no	6
[23]	yes	yes	yes	NC	yes	yes	no	yes	no	yes	7
[24]	yes	yes	yes	NC	yes	yes	no	yes	no	yes	7
[25]	yes	yes	no	NC	yes	yes	yes	yes	no	yes	7
[26]	no	yes	yes	No	yes	yes	no	yes	NA	yes	6

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
