# Peer review of "Therapeutic Approaches in Adult Primary Spinal Cord Astrocytoma: A Systematic Review"

_cancers, 2022, doi:10.3390/cancers14051292_

Round 1

Reviewer 1 Report

The authors report the results of a PubMed search on spinal astrocytoma in adults. Among 1197 citations, they identified 16 publications meeting the inclusion criteria. The methodology of the analysis is appropriate. The results are reported unbiased. However, the reported endpoints of the published series were too heterogeneous for a meaningful meta-analysis. Despite this disappointing result, the paper is of interest for the physicians who need an overview over published date while treating patients with rare intramedullary astrocytoma. Moreover, the authors emphasize the need of standardized clinical registries and trials for the development of treatment recommendations. I have no major concerns. Minor comments: (i) Line 162+: The authors should revise the phrase because the number of items does not match the number of data items. (ii) Table 2 is very busy; perhaps the authors could condense the content in an effort to improve readability. (iii) Table 3: The authors should include the Risk-of-bias assessment total score for each publication. (iv) The supplemental table seems to have redundant columns for example on surgery and radiotherapy. Please revise and include a detailed legend in English.

Author Response

We thank the reviewer for her/his insightful comments.

Minor comments:

About minor concerns, we have addressed them in detail as reported below.

(i) Line 162+: The authors should revise the phrase because the number of items does not match the number of data items.

We thank the reviewer for spotting this inaccuracy in the main text. The period was “As shown in Supplementary Table 1, gliomas spinal site in cervical thoracic and lumbar regions ranged from 0% (n = 0/3, [17]) to 70% (n = 7/10, [25]), from 20% (n = 2/10, [25]) to 67% (n = 14/21, [26]) and from 0 (n = 0/13, [10]; n = 0/24, [11]; n = 0/22, [15], n =  0/6, [18]; n = 0/5, [19]; n = 0/25, [21]; n = 0/6, [22]; n = 0/13, [23]; n = 0/21, [23]) to 41% (n = 5/12, [14]) respectively.”

We have now changed to (in bold the modification): “As shown in Supplementary Table 1, gliomas spinal site in cervical thoracic and lumbar regions ranged from 0% (n = 0/3, [17]) to 70% (n = 7/10, [25]), from 20% (n = 2/10, [25]) to 77% (n = 10/13, [10]) and from 0 (n = 0/13, [10]; n = 0/24, [11]; n = 0/22, [15], n =  0/6, [18]; n = 0/5, [19]; n = 0/25, [21]; n = 0/6, [22]; n = 0/13, [23]; n = 0/21, [26]) to 41% (n = 5/12, [14]) respectively.”

(ii) Table 2 is very busy; perhaps the authors could condense the content in an effort to improve readability.

We apologize for the hardly readable Table 2. We have now revised the Table, and made it more simplified.

(iii) Table 3: The authors should include the Risk-of-bias assessment total score for each publication.

As the reviewer suggested, we have added a column named “total score” to Table 3 (Risk-of-bias assessment).

(iv) The supplemental table seems to have redundant columns for example on surgery and radiotherapy. Please revise and include a detailed legend in English.

We thank the reviewer. We have now revised the Table and added a detailed Legend on Main Manuscript.

We have also corrected some typing errors in the main text, that now resulted as marked up by “Track Changes” function.

Reviewer 2 Report

The manuscript is well written and is affording a large cohort of a rare low or high grade astrocytoma. Nevertheless, the data are enlarging the knowledge we have on those adult gliomas. I will only ask for a restructured introduction wher the authors should add more previous data and description of the usual characteristics present in low and high grade gliomas of the adults. The introduction is a bit short to introduce the potential specifities of the spinal cord locations.       

Reviewer 3 Report

The review of this rare spinal cord tumor is exhaustive, helped by the fact that the clinical knowledge is really limited .I agree with the authors that a cooperative study about this pathology should be done. Interesting how also the molecular profile remains behind the brain tumor experience.

This paper is a useful summary of today knowledge in approaching this pathology.

Author Response

We really thank the reviewer.